# Missed Cystic Fibrosis Newborn Screening Cases due to Immunoreactive Trypsinogen Levels below Program Cutoffs: A National Survey of Risk Factors

**DOI:** 10.3390/ijns8040058

**Published:** 2022-10-27

**Authors:** Martin Kharrazi, Charlene Sacramento, Anne Marie Comeau, Jaime E. Hale, Michele Caggana, Denise M. Kay, Rachel Lee, Brendan Reilly, John D. Thompson, Samya Z. Nasr, Mary Kleyn, Gary Hoffman, Mei W. Baker, Colleen Clarke, Cheryl L. Harris, M. Christine Dorley, Hilary Fryman, Ankit Sutaria, Amy Hietala, Holly Winslow, Holly Richards, Bradford L. Therrell

**Affiliations:** 1California Department of Public Health, Richmond, CA 94804, USA; 2Sequoia Foundation, La Jolla, CA 92037, USA; 3New England Newborn Screening Program, UMass Chan Medical School, Worcester, MA 01605, USA; 4New York State Newborn Screening Program, Wadsworth Center, New York State Department of Health, Albany, NY 12208, USA; 5Texas Department of State Health Services, Austin, TX 78756, USA; 6Washington State Department of Health, Public Health Laboratories, Newborn Screening Program, Shoreline, WA 98155, USA; 7Michigan Medicine, University of Michigan, Ann Arbor, MI 48109, USA; 8Michigan Department of Health and Human Services, Lansing, MI 48913, USA; 9Wisconsin Newborn Screening Laboratory at Wisconsin State Laboratory of Hygiene, University of Wisconsin School of Medicine and Public Health, Madison, WI 53706, USA; 10Louisiana Office of Public Health, Baton Rouge, LA 70802, USA; 11Tennessee Department of Health, Newborn Screening Laboratory and Follow-Up Program, Nashville, TN 37216, USA; 12Georgia Department of Public Health, Atlanta, GA 30303, USA; 13Minnesota Department of Health, Public Health Laboratory, Newborn Screening, St. Paul, MN 55164, USA; 14Maine Newborn Screening Program, Augusta, ME 04333, USA; 15National Newborn Screening and Global Resource Center, Austin, TX 78759, USA; 16Department of Pediatrics, The University of Texas Health Science Center at San Antonio, San Antonio, TX 78249, USA

**Keywords:** cystic fibrosis, CF, newborn screening, missed cases, immunoreactive trypsinogen, IRT

## Abstract

Testing immunoreactive trypsinogen (IRT) is the first step in cystic fibrosis (CF) newborn screening. While high IRT is associated with CF, some cases are missed. This survey aimed to find factors associated with missed CF cases due to IRT levels below program cutoffs. Twenty-nine states responded to a U.S-wide survey and 13 supplied program-related data for low IRT false screen negative cases (CFFN) and CF true screen positive cases (CFTP) for analysis. Rates of missed CF cases and odds ratios were derived for each factor in CFFNs, and two CFFN subgroups, IRT above (“high”) and below (“low”) the CFFN median (39 ng/mL) compared to CFTPs for this entire sample set. Factors associated with “high” CFFN subgroup were Black race, higher IRT cutoff, fixed IRT cutoff, genotypes without two known CF-causing variants, and meconium ileus. Factors associated with “low” CFFN subgroup were older age at specimen collection, Saturday birth, hotter season of newborn dried blood spot collection, maximum ≥ 3 days laboratories could be closed, preterm birth, and formula feeding newborns. Lowering IRT cutoffs may reduce “high” IRT CFFNs. Addressing hospital and laboratory factors (like training staff in collection of blood spots, using insulated containers during transport and reducing consecutive days screening laboratories are closed) may reduce “low” IRT CFFNs.

## 1. Introduction

A cystic fibrosis (CF) diagnosis occurring after two months of age is associated with poorer health outcomes, including height below the 10th percentile, higher prevalence of *Pseudomonas aeruginosa* infection and diminished lung function [1]. In order to make an early diagnosis, some form of newborn screening (NBS) for CF was implemented in the USA at various times. Colorado began with an “IRT only” algorithm in 1982 [2], followed by Wisconsin which introduced a second-tier test for the most common *CFTR* variant in order to enhance screening specificity in 1994 [3]. Massachusetts introduced the use of a multi-variant *CFTR* panel to maintain specificity while enhancing sensitivity for heterogeneous populations in 1999 [4]. The Cystic Fibrosis Foundation voted in 2008 in favor of universal CF NBS, which was achieved by 2010 (https://www.cff.org/About-Us/Media-Center/Press-Releases/All-Fifty-States-to-Screen-Newborns-for-Cystic-Fibrosis-by-2010/, accessed on 26 November 2019). The first step in all CF NBS algorithms currently is measurement of immunoreactive trypsinogen (IRT) in newborn dried blood spots [5]. IRT is a pancreatic protein elevated in serum by about ten-fold in CF cases during the first 2–3 months of life, thereafter falling to levels found in the overall live birth population [6]. It has been estimated that 5% of CF cases are missed at this first critical step because IRT levels were reported to be below pre-set program IRT cutoffs [7].

Factors that can result in missed CF cases have been described broadly [8], but there is sparse literature on factors that are related specifically to low IRT false screen negative cases (CFFN). NBS programs with higher IRT cutoffs (e.g., above 99th percentile) have lower screening sensitivity than those with lower cutoffs [9]. Even at the highest levels of sensitivity, IRT below cutoffs remains the main reason for missed CF cases [9,10,11].

In an attempt to identify other factors related to CFFN, investigators have examined the overall population distribution of IRT (usually median and 95% percentile) in all newborns [9]. Lower IRT levels can occur when testing is delayed and in summer versus winter, both thought to be due to degradation of IRT when blood spots are kept at length in warm temperatures [8,9] and in newborns who are at an older age at blood spot collection [12]. Higher IRT levels have been observed among newborns of Black women compared to those of White women [13], babies with birth weight < 1500 g [9], and babies who experience stress during parturition, placement in the neonatal intensive care unit, or transfusion [8]. It is also known that IRT assay kit lots show variability [9]. Some NBS programs have attempted to use floating cutoff values based on a daily or weekly percentile versus set value cutoffs to adjust for seasonal and kit variability [9].

While most state NBS programs have reported CFFN cases [7], numbers have been too small for even the largest programs to conduct a well-powered and meaningful analysis of possible causes. It is the aim of this study to identify characteristics of CFFN cases that differ from identified CF true screen positive cases (CFTP). It serves as a first exploratory step in addressing the call in 2012 by Therrell and colleagues [7] for a nationwide effort. In this analysis, we focus on a wide array of readily available factors recorded by NBS programs in the USA that may reveal options to lower CFFN rates. These include program-, biologic-, health-, demographic-, time-, hospital- and laboratory-related risk factors.

## 2. Materials and Methods

NBS programs in 50 states and the District of Columbia were approached in 2012 by email, telephone and in-person and over several years to participate in this project. Twenty-nine states provided data from their newborn screening program for at least one- of the four-part survey. The survey was conducted in two phases. The first phase included births screened using CF algorithms in place on 31 December 2010 and backward to the algorithm implementation date. The second phase extended this period to 31 December 2012, allowing in both phases for at least a two-year period for states to learn of any missed CF cases. This design permitted different starting points but the same 31 December 2012 end date for all but two states that ended on 31 December 2010.

The survey was comprised of Appendix A (see online supplement for forms and completion instructions (Appendix A)). We collected information about: (i) state program attributes, (ii) CFFN cases due to IRT below program cutoffs during the first step(s) of the CF algorithm, (iii) CFTP cases identified by the newborn screening program (plus the few missed cases from second or third steps, like *CFTR* variants not on the state mutation panel), and (iv) all newborns screened by the program (or live births as a substitution in one state, LB). In addition, we conducted phone interviews/meetings with states to collect clarifying and Appendix A about lab and follow-up procedures (or collected via email if not available to meet). We relied on states to define CF cases. An examination of 55 CFFN cases found that 52 had one or more of the following: a sweat chloride test ≥ 60 mmol/L, two known CF-causing variants according to CFTR2 (https://cftr2.org/, accessed on 30 December 2019), or a family history of CF, Of the remaining 3 CFFN cases, one had a borderline sweat chloride test (57 mmol/L) and 1 known CF-causing variant and 1 variable clinical consequence variant, and two did not have sufficient data available to make a determination.

Since no HIPAA (Health Insurance Portability and Accountability Act of 1996) identifiers were used, this project was considered exempt by the California State Health and Human Services Agency Committee for the Protection of Human Subjects. Four states required formal data requests, collaborative research agreements, or institutional review board approvals.

Study variables are listed in Table 1 and grouped into demographic, timing and program, CF algorithm, and biologic and health factors. Table 1 also shows the number of states and study groups included in analysis of each factor; not all 13 states collected or provided data on all factors. The LB and CFTP study group data were provided by each state at a summary level, i.e., numbers in response categories (e.g., Black race and month specimen collected) or medians (e.g., age at specimen collection and IRT value). Individual-level data were provided for the CFFN study group but reported in categories (e.g., preterm birth and term birth instead of gestational duration in weeks and days) when there was concern around maintaining subject anonymity. Four lab-related variables were defined at the program level (Table 1). This purposeful approach to data collection was required to encourage programs’ participation and to minimize the amount of individual-level data sharing. Further, we assured programs that only aggregated results would be presented so no specific state could be identified.

In the analysis, characteristics of CFFN cases were compared to those of CFTP cases (and the LB cohort screened during the study period). After summing all state numbers for each study group by response categories, the rate of missed CF cases due to low IRT (CFFN_i_/(CFTP_i_ + CFFN_i_)) was calculated for each response category_(i)_ of categorical variables (e.g., Black, White, Other race; each day of the week of birth) and compared for each risk factor. These rates of missed CF cases due to low IRT should be seen as minimum rates. Although the study design allowed for a minimum of 2 years of follow up, not all missed cases will have been identified by programs in that timeframe. Despite the undercount, cross category comparisons should be relatively accurate and interpretable. For some categories, however, we note when this assumption is unlikely (e.g., meconium ileus identified in the newborn).

To measure the association between variable responses in CFFN and CFTP groups, odds ratios (i.e., odds of response in CFFNs vs. the odds of response in CFTPs) and 90% confidence intervals were used (Odds ratio—Confidence Interval—Select Statistical Consultants (https://select-statistics.co.uk/calculators/confidence-interval-calculator-odds-ratio/, accessed on various dates 15 January 2022–30 June 2022). A more liberal 90% (vs. 95%) confidence interval was chosen given the early stage and exploratory nature of this investigation. The selection of odds ratios versus rate ratios was purposeful to preserve the nature of how the data were collected, to allow for CFFN subgroup analyses, and in recognition that not all missed cases were likely to have been found and reported.

IRT values, newborn age at blood collection and age at time of IRT testing were compared across study groups using the average of state-specific medians, weighted by the number in each state. CFFN cases were also stratified according to initial specimen IRT values below and above the median IRT value of CFFNs (i.e., <40 ng/mL, which includes the median and ≥40 ng/mL) to distinguish two subgroups, (i) CFFNs that would have been missed by states using even the lowest IRT cutoff and (ii) CFFNs with IRT values closer to current program cutoffs. The aggregate nature of the data only permitted bivariate analyses of risk factors, i.e., one variable at a time with no covariate adjustment. By design, individual-level risk factors were less likely to be confounded than state-level risk factors due to consistencies in screening and follow up methods within states for all study groups.

## 3. Results

### 3.1. Program Recruitment

Twenty-two states did not participate in the survey. Reasons included lack of staff time (4), did not collect requested data or did not collect it consistently (2), data was not credible or no longer available (2), no response to investigators (7) and other reasons (7). Of the 29 states providing data, 16 were later excluded from analysis because three critical sections 1–3 of the survey were not completed (15) or the program follow up of missed cases was deemed inadequate based on legal, regulatory and procedural criteria (1). This left data from 13 states eligible for the analyses. Ten of these states were 1-specimen states and three were 2-specimen states (i.e., NBS programs requiring a second specimen at 1–2 weeks of age to generate results). Together, these states covered 47.1% of U.S. births according to 2013 CDC figures.

### 3.2. Study Subjects

There was a total of 63 CFFNs, 2019 CFTPs and over 11 million LBs reported in the 13 participating states. Two CFFNs had their specimens collected at over 6 months of age—when an IRT value would no longer be elevated in CF cases—and these samples were dropped from further analyses in this study. Two of the 13 states reported 0 CFFN cases. The range of IRT cutoffs used by the 13 states was 0.95–0.998 percentile. The percent of CF cases missed because of an IRT value below program cutoffs was 2.93%. The overall CF prevalence at birth was 1 case per 5402 live births.

### 3.3. IRT Values

State-weighted average median IRT values for the three main study groups were, as expected, highest among CFTPs, lowest among LBs, and intermediate among CFFNs. Figure 1 shows state medians for IRT separately for one- and two-specimen states; average median IRT values for initial specimens were quite similar across one- and two-specimen states for the LB and CFTP groups, but higher for CFFNs in two-specimen states indicative of their IRT cut off values usually being higher than in one-specimen states. The range of IRT values in CFFN cases was 8.0–103.5 ng/mL (median: 39 ng/mL, ~0.80 percentile of the screened population) (Figure 2). Half of all CFFNs had IRT values well below commonly used cutoffs (≥0.95 percentile, ~52 ng/mL or higher).

### 3.4. Age at CF Diagnosis

In the 12 states that collected date of CF diagnosis, CFFNs had a large delay in diagnosis versus CFTPs (Figure 3). Only 46% of CFFNs versus 84% of CFTPs were diagnosed before 57 days of age, whereas 37% of CFFNs versus 4% of CFTPs were diagnosed after 158 days.

### 3.5. Risk Factors

In the presentation of the following factors, the main comparisons made are between CFFN and CFTP groups. Summary data for categorical responses can be found for LB, CFTP and CFFN groups in Table 2. For each category, the rate of missed CF cases due to having IRT values below program cutoffs is presented (Table 2). Odds ratios and 90% confidence intervals in CFFN, and in subgroups (CFFN ≥ 40 ng/mL and CFFN < 40 ng/mL) compared to CFTP are in Table 3 for these factors. Confidence intervals for odds ratios ≥ 2.0 or ≤0.5 are included to show the spread in the point estimate.

#### 3.5.1. Demographic Factors

##### Sex

The rate of missed CF cases due to low IRT was the same for males and females. The odds of being a missed female were similar across all study groups, including CFFNs with IRT < 40 ng/mL or ≥40 ng/mL).

##### Race

Black newborns had a higher rate of missed CF cases due to low IRT (9.21%) than White newborns (2.90%) and other (4.20%) races. Black newborns were overrepresented among CFFN cases (OR = 3.4, 90% CI 1.7, 6.8), with the association being strongest in CFFNs with IRT ≥ 40 ng/mL (OR = 4.2, 90% CI 1.7, 10.6).

##### Ethnicity

Hispanic newborns had a lower rate of missed CF cases due to low IRT (2.60%) than non-Hispanics (3.78%). Hispanic newborns were underrepresented among CFFN cases (OR = 0.7, 90% CI 0.4, 1.3), with the association being entirely due to CFFNs with IRT ≥ 40 ng/mL (OR = 0.2, 90% CI (0.0, 0.98).

#### 3.5.2. Timing and Program Factors

##### Age at Specimen Collection and Testing

The weighted average median age for all study groups across one- and two-specimen states at specimen collection and lab testing is presented in Figure 4, Figure 5 and Figure 6. CFFNs with IRT levels < 40 ng/mL consistently had a higher median age at specimen collection than all other groups. The weighted average median age at testing was similar across the study groups, except for Specimen 2 in two-specimen states where the median age at testing was 19.0 days in CFFNs with IRT < 40 ng/mL compared to 16.8 days in CFTPs, a difference in medians of 2.2 days. In one-specimen states, CFFNs with IRT < 40 ng/mL had their specimens collected at a median age of 41.6 h, while in CFTPs the median was 39.8 h, a difference in medians of 1.8 h. For two-specimen states, CFFNs with IRT < 40 ng/mL had their first specimen collected at a median age of 44.2 h, while in CFTPs the median was 30.3 h, a difference in medians of 13.9 h, while for their second specimen the median ages at collection were 385.0 and 301.4 h, respectively, for a difference of 83.6 h.

##### Day of the Week of Birth

The rate of missed CF cases due to low IRT was 1.63% on Sunday, 1.59% on Tuesday, and 4.76% on Saturday. When compared to other days of the week, there were lower odds ratios on Tuesday (OR = 0.4, 90% CI 0.2, 0.99), suggestively lower odds ratios on Sunday (OR = 0.6, 90% CI 0.2, 1.5), and a higher odds ratio on Saturday (OR = 2.0, 90% CI 1.2, 3.6). The Saturday findings were magnified in CFFNs with IRT <40 ng/mL (OR = 2.5, 90% CI 1.2, 5.3).

##### Season Specimen Collected

Due to the number of CFFNs, months were collapsed into seasons for analysis. The rate of missed CF cases due to low IRT was 2.76% in Winter (December, January, February), 2.03% in Spring (March, April, May), 3.27% in Summer (June, July August), and 3.47% in Fall (September, October, November). The odds of specimen collections were lowest in the Winter and Spring in CFFNs compared to CFTPs and highest in the Summer and Fall. These results were only found in CFFNs with IRT < 40 ng/mL (OR = 2.0, 90% CI 1.1, 4.0).

##### Maximum Consecutive Days Lab Closed

The rate of missed CF cases due to low IRT was higher in state laboratories with maximum 3 or more consecutive days closed (3.31%) versus fewer days (2.56%). These results were due to CFFNs with IRT < 40 ng/mL (OR = 2.0, 90% CI 1.1, 3.8).

##### Program Type

A larger percentage of CFFNs were missed due to low IRT when born in two- (3.42%) versus one- (2.82%) specimen states. The odds ratios suggest that this association was driven by CFFNs with IRT ≥ 40 ng/mL.

#### 3.5.3. CF Algorithm Factors

##### IRT Cutoff Type

The rate of missed CF cases due to low IRT was 3.48% in states with a fixed cutoff and 2.36% in states with a floating cutoff. The odds ratios indicate that this was due mostly to higher odds in CFFNs with IRT ≥ 4 0 ng/mL (OR = 2.0, 90% CI 1.1, 3.8).

##### IRT Cutoff Level

Rates of missed CF cases due to low IRT ranged from 2.54% to 4.55% when cutoffs increased from <0.96 percentile to ≥0.99 percentile, respectively. When dichotomized at ≥0.96 and <0.96 percentile, the odds ratios were uniquely higher in CFFNs with IRT ≥ 40 ng/mL (OR = 1.7, 90% CI 0.9, 3.2), which follows expectations.

#### 3.5.4. Biologic and Health Factors

##### *CFTR* Genotype

There was a striking difference in the rates of missed CF cases due to low IRT when categorized by genotype: 2.04% were missed in newborns with two known CF-causing variants, 5.06% were missed in newborns with other genotypes, and 2.88% were missed in newborns tested but with both variants unidentified. The odds of being a CFFN with other genotypes (i.e., without two known CF-causing variants) were higher compared to that for a CFTP (OR = 2.5, 90% CI 1.6, 3.9), and consequentially lower in CFFNs with two known CF-causing variants (OR = 0.4, 90% CI 0.3, 0.7). These associations were predominantly driven by CFFNs with IRT ≥ 40 ng/mL (OR = 4.4, 90% CI 2.2, 8.7) and (OR = 0.2, 90% CI 0.1, 0.5), respectively. The CFFN/CFTP odds ratios were unremarkable for newborns tested but with both variants unidentified.

##### Birth Weight

The rate of missed CF cases due to low IRT was 3.69% in low birth weight (<2500 g) infants and 2.89% in higher weight infants. The odds of low birth weight were somewhat higher in CFFNs than in CFTPs, primarily due to CFFNs with IRT < 40 ng/mL (OR = 1.6, 90% CI 0.7, 3.7), as was found with a stronger association in preterm births (described below).

##### Gestational Duration

There was a much higher rate of missed CF cases due to low IRT in newborns born preterm (<37 weeks gestation, 7.41%) than in those born at term (≥37 weeks, 2.59%). The odds of being preterm were higher in CFFNs than in CFTPs (OR = 3.0, 90% CI 1.1, 8.2). This was almost entirely due to CFFNs with IRT < 40 ng/mL (OR = 7.5, 90% CI 1.9, 29.5).

##### Infant Feeding

The rate of missed CF cases due to low IRT was higher in newborns fed formula only or formula with breast milk (3.90%) than in those fed breast milk only (1.94%). The odds of any formula use were higher in CFFNs than in CFTPs (OR = 2.0, 90% CI 1.01, 4.1). This association was exclusively found in CFFNs with IRT < 40 ng/mL (OR = 12.3, 90% CI 2.2, 69.1).

##### Meconium Ileus

The rate of missed CF cases due to low IRT for newborns with meconium ileus was higher (6.11%) than in those without meconium ileus (2.75%). The odds of meconium ileus being present in CFFNs were higher than in CFTPs (OR = 2.3, 90% CI 1.2, 4.2). This association was mostly due to CFFNs with IRT ≥ 40 ng/mL (OR = 2.7, 90% CI 1.3, 5.9).

## 4. Discussion

This is the first multi-state study of missed CF cases due to IRT levels below program cutoffs in the USA. The states chosen for inclusion needed to have the ability to identify missed CF cases. The cases derive from over 11 million live births with a collective CF prevalence at birth of at least 1 case per 5402 live births.

A broad array of factors was associated with CFFNs. Different types of risk factors were associated with each of the two CFFN subgroups (i.e., the half with ”higher” and “lower” IRT levels). We discuss the risk factors that appear to be associated with potentially mutable practices separately from those that appear to be associated with newborn characteristics.

### 4.1. Factors Associated with Timing and Practices

For the CFFN subgroup with “higher” IRT values, the strongest and most significant mutable factors were higher program IRT cutoff and fixed IRT cutoff. Use of a lower IRT cutoff and a floating cutoff may be helpful in reducing the numbers in this subgroup of CFFNs. Changing cutoff levels must be done carefully so that false positive screens are maintained at reasonable levels. To keep cutoff levels relatively stable, some programs have recently refined their methods to exclude certain specimens with characteristics known to be associated with higher IRT values (e.g., those that were collected within 48 h of a transfusion and those collected from very low birth weight infants) from the calculation of the percentile used for the cutoff [14]. Because many states that utilize a floating cutoff in this study are the same states that have the lowest IRT cutoffs, it is hard to be confident that a floating cutoff by itself is driving this finding. The benefits of floating versus fixed cutoffs deserve further study.

For the CFFN subgroup with “lower” IRT values, the strongest and most significant factors were a higher median newborn age at specimen collection, birth on a Saturday, hotter season of newborn dried blood spot collection, and a maximum of three or more days laboratories could be closed. Screening newborns at an earlier age may come with benefits for CF NBS. The recommendation of screening newborns aged between 24 and 48 h that occurred after the collection of data for this study in the USA [15] may have helped lower the rate of CFFNs in as much as this recommendation was successfully implemented. This recommendation may not be reasonable for areas of the world where collection of dried blood spots is not birth hospital-based and occurs at later ages [16]. One possible explanation for the higher CFFN versus CFTP odds for Saturday births may be that hospital personnel who are less experienced with newborn screening procedures working on weekends. This explanation is supported by a higher median age at specimen collection, and by Sunday and Tuesday births having the lowest percentages of missed cases. A recent study found that low quality dried blood spots were associated with CFFNs [17]. Evaluation and training of weekend staff may be useful here. Hotter seasons and three or more days newborn screening laboratories can be closed may be associated with the higher likelihood of IRT degradation. Steps to reduce IRT degradation could be to provide some type of insulated specimen container for newborn specimens in collection facilities and during transport to screening laboratories and reducing the period that screening laboratories are closed because of holidays and other factors. Since the data collection for this paper, there have been significant changes to two of these timing factors nationwide: programs have taken steps to prevent specimen batching by birth hospitals and more programs require a maximum of two days in which CF screening operations are closed (https://www.newsteps.org/resources/toolkits/timeliness-toolkit-expanding-newborn-screening-services, accessed on 15 August 2022).

### 4.2. Factors Associated with Newborn Characteristics

For the CFFN subgroup with “higher” IRT values, the strongest and most significant factors were presence of meconium ileus, Black race, and genotypes with less than two known CF-causing variants. The association of meconium ileus may be due to a greater likelihood of identifying CF cases with meconium ileus in the hospital even when the IRT is below cutoff levels, and lower IRT values being reported generally for CF cases with meconium ileus in some but not all states [7]. The meconium ileus finding in CFFNs is of little significance to newborn screening because nearly all newborns with the condition are CF identified shortly after birth and should be at heightened suspicion for CF regardless of screening results. The high rate associated with Black race warrants further study. There are reports of Black infants having higher IRT levels [7,13] so our finding of a higher odds in CFFNs compared to CFTPs is contrary to that expectation. Higher odds were found in newborns with genotypes not comprised of two known CF-causing variants (i.e., only one known CF-causing variant and a second variant with varying clinical consequence, not identified by the test used or not yet evaluated by CFTR2). Such newborns have shown to have lower IRT levels [18]

For the CFFN subgroup with “lower” IRT values, the strongest and most significant factors were preterm birth, and formula feeding of newborns. Preterm births (and stressed births, generally) have been reported to have babies with higher IRT levels [8] so this finding of a higher odds in CFFNs compared to CFTPs is contrary to that expectation. To the best of our knowledge, formula or mixed breast and formula feeding have not been reported to be related to CFFNs or to a lower IRT in general.

An early finding of this study was two CFFN cases that were missed because of advanced age at specimen collection, and subsequently excluded from the analysis. Despite widespread educational efforts on timing of sample collection by programs, late collection is still an issue for certain populations (e.g., homebirths and parent refusals for screening until symptoms appear). Programs should develop and enact procedures to address newborns expected a priori to have low IRT levels. One possible solution for CF NBS programs with a DNA testing component could be to go straight to DNA testing of the screening specimens of these infants, in lieu of an IRT-first approach.

### 4.3. Practice Recommendations

The findings from this study may be useful in lowering the number of missed cases due to low IRT. As suggested by other CF newborn screening researchers [5], lowering the IRT cutoff, possibly by using IRT/DNA-based algorithms, will be effective in identifying CFFNs with “higher” IRT levels. Approaches to protect dried blood spot specimens from IRT degradation through improved specimen handling and training of hospital staff in newborn screening specimen collection may be effective in identifying CFFNs with lower IRT levels. Newborn screening laboratories should continue to explore ways to stay open with fewer stretches of time being closed, or possibly receiving specimens 6–7 days per week and keeping them in refrigerated temporary storage until testing can begin. This recommendation may have benefits beyond CF to maintain levels of other analytes with degradation issues used for screening disorders, like galactosemia [19,20]. Initiatives to lessen transport time from hospitals to screening labs in the years after data collection for this study may have at least partially addressed this factor (https://www.newsteps.org/resources/toolkits/timeliness-toolkit-expanding-newborn-screening-services, accessed on 20 August 2022).

### 4.4. Study Strengths and Weaknesses

Insight into CFFN risk factors was improved by separate analysis of CFFNs below and above the median IRT level (i.e., <40 ng/mL, which is below any current program cutoffs, and ≥40 ng/mL), respectively). However, this study was limited by the collection of summary-level state data, especially among CFTPs, because it does not allow for multivariable analyses that may be able to disentangle effects among the risk factors identified. Even though we did not require states to use a standardized definition of CF [21], our examination of CFFN cases suggested they did. There were challenges in recruiting or inclusion of all 51 NBS programs due to (i) non-response, (ii) lack of usable data, (iii) difficulty accessing data, (iv) insufficient staff time, and (v) no routine and long-term follow up of false screen negative cases. Investigations with high-quality, individual-level data from a very large cohort of CF cases that include excellent follow up of missed cases are needed to verify these findings. Such studies could also help provide more data on factors that have been found to have elevated IRT distributions in births but also have higher rates of CFFN cases, like Black race, very low birth weight and preterm birth.

## 5. Conclusions

This investigation found multiple risk factors associated with missed CF cases due to IRT levels below newborn screening program cutoffs. These results suggest that the number of missed cases can be reduced by lowering program IRT cutoff levels, and by improving various hospital, specimen transport and laboratory practices that likely lead to IRT degradation.

## Figures and Tables

**Figure 1 IJNS-08-00058-f001:**
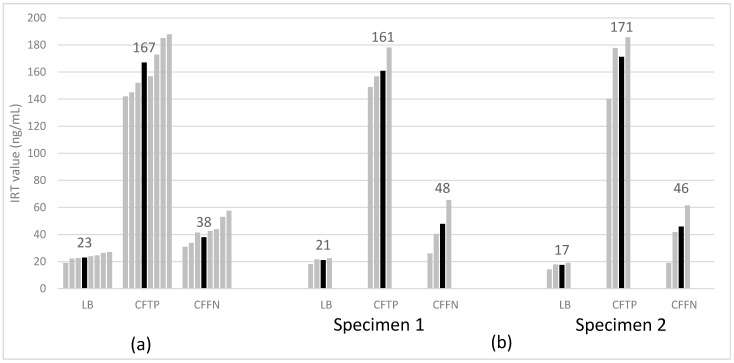
State median (gray bars) and weighted average median (black bars) IRT value for study groups in 7 one-specimen (**a**) and 3 two-specimen (**b**) states. CFFN, false screen negative CF cases due to IRT below program cutoffs; CFTP, true screen positive CF infants; IRT, immunoreactive trypsinogen; LB, all screened newborns.

**Figure 2 IJNS-08-00058-f002:**
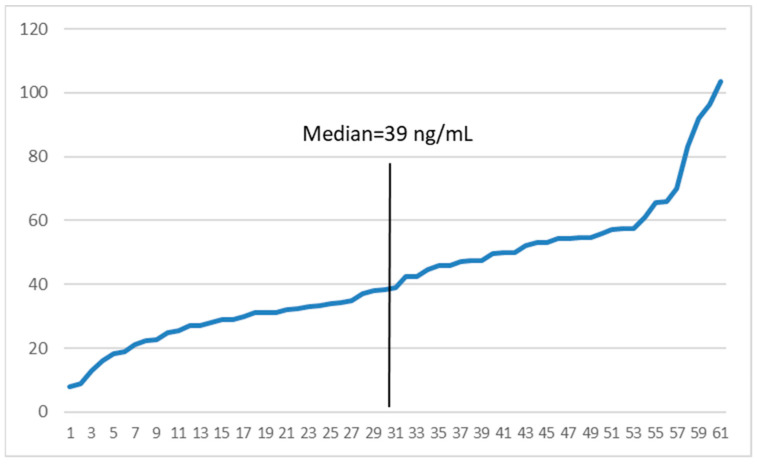
IRT values (ng/mL) and median for 61 CFFN cases in 11 states (sorted lowest to highest). CFFN, false screen negative CF cases due to IRT below program cutoffs; IRT, immunoreactive trypsinogen.

**Figure 3 IJNS-08-00058-f003:**
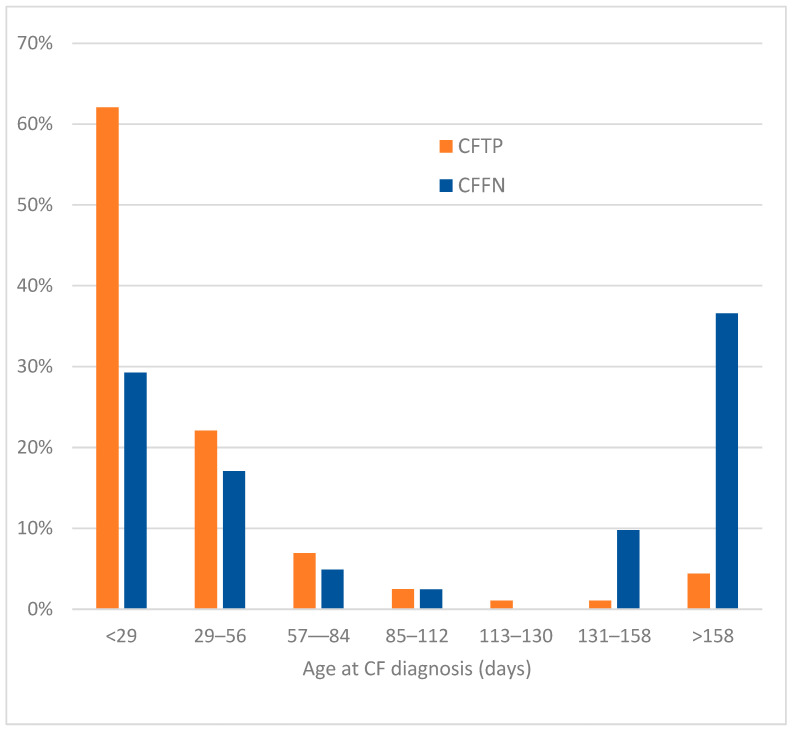
Percentage of 41 CFFN and 1545 CFTP cases by age at CF diagnosis in 12 states.

**Figure 4 IJNS-08-00058-f004:**
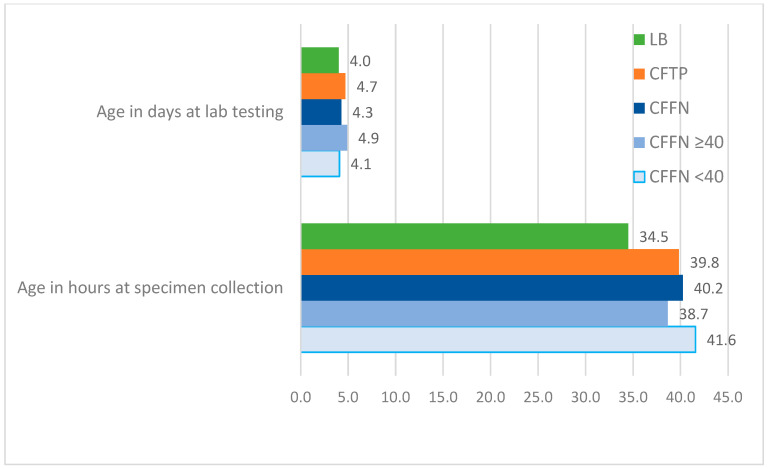
Weighted average median age at specimen collection and laboratory testing for all study groups in 6 one-specimen states.

**Figure 5 IJNS-08-00058-f005:**
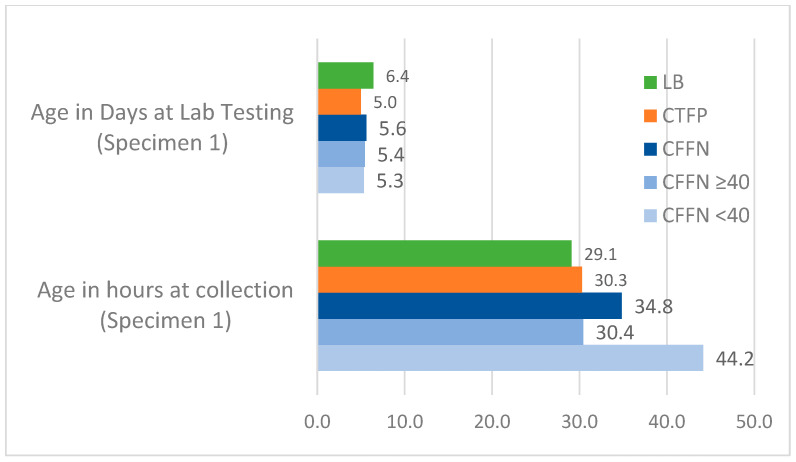
Weighted average median age at specimen collection and laboratory testing for all study groups in 3 two-specimen states: Specimen 1. (Median ages for CFFNs with IRT ≥40 ng/mL based on 2 states).

**Figure 6 IJNS-08-00058-f006:**
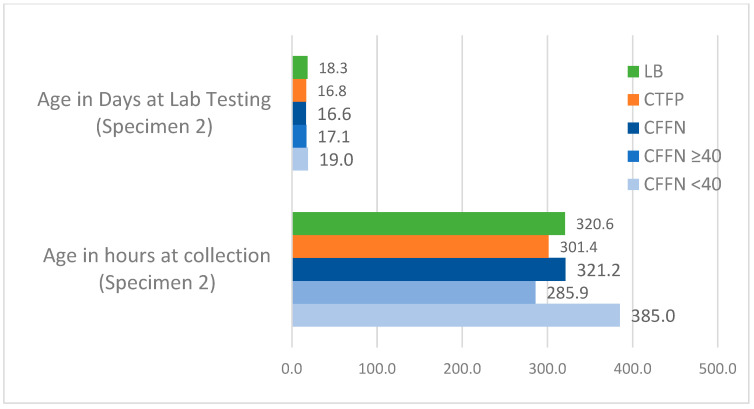
Weighted average median age at specimen collection and laboratory testing for all study groups in 3 two-specimen states: Specimen 2. (Median ages for CFFNs with IRT ≥40 ng/mL based on 2 states).

**Table 1 IJNS-08-00058-t001:** Study variables by inclusion and exclusion criteria, number of states and response categories used in main analyses.

Variable	Inclusion	Exclusion ^1^	Number of States	Response Categories
Demographic Factors				
Sex	All states ^2^	Subjects with sex missing or unknown	Summed for 13 states	% female, male
Race	States providing data for all study groups ^2^	2 States not collecting data; Subjects with race missing or unknown	Summed for 11 states	% White, Black, Other
Ethnicity	States providing data for all study groups ^2^	2 States not collecting data; subjects with ethnicity missing or unknown	Summed for 11 states	% Hispanic
Timing and Program Factors				
Median newborn age at specimen collection and IRT testing	States providing data; subjects had data for both variables	2 States not providing one or both data items for all study groups; 2 States with zero CFFNs	9 states stratified by 1- and 2-specimen programs	Average median age weighted by state birth counts and number of CFTPs and CFFNs
Day of week birth occurred	States providing data for all study groups	2 States with missing data for CFFNs or CFTPs	Summed for 11 states	% born on different days of the week
Season of Specimen Collection	States providing data on month of specimen collection for all study groups	2 States with missing data for CFFNs or CFTPs	Summed for 11 states	% specimens collected in different seasons of the year
Maximum consecutive days lab closed ^3^	All states	none	Summed for 13 states	% in states with a lab that can be closed for a maximum of 3 or more or less than 3 days
Program type ^3^	All states	none	Summed for 13 states	% in states screening 1- or 2- specimens
CF Algorithm Factors				
Median IRT value	States providing data for all study groups	3 States with missing data for CFTPs and LBs	10 states stratified by 1- and 2-specimen programs	Average median IRT weighted by the number of LBs, CFTPs and CFFNs
IRT cutoff type ^3^	All states	none	Summed for 13 states	% in states using a fixed or floating IRT cutoff
IRT cutoff level ^3^	All states	none	Summed for 13 states	% in states with an IRT cutoff of <0.96, ≥0.96, ≥0.97, ≥0.98, ≥0.99 percentiles
Biologic and Health Factors				
*CFTR* genotype ^4^	States providing data for both CFTP and CFFN study groups	2 States not collecting data; subjects not undergoing genotype testing	Summed for 11 states	% in ranked categories based on CFTR2 ^5^: 1. Both variants unidentified 2. Both variants known CF-causing 3. All other genotype combinations
Birth weight	All states ^2^	Subjects with birth weight missing or unknown	Summed for 13 states	% <2500, ≥2500 g
Gestational duration	States providing data for all study groups ^2^	5 States not collecting data; 2 States with a high proportion of missing data; subjects with gestational age missing or unknown	Summed for 6 states	% <37, ≥37 completed weeks
Infant feeding	States providing data for all study groups	7 states not collecting data; subjects with feeding data unknown or missing	Summed for 6 states	% breast only, any formula use
Meconium ileus	States providing data for both CFTP and CFFN study groups	4 States not collecting data	Summed for 9 states	% meconium ileus present, not indicated

Abbreviations: CFFN, false screen negative CF cases due to IRT below program cutoffs; CFTP, true screen positive CF infants; IRT, immunoreactive trypsinogen; LB, all screened newborns. ^1^ Two CFFNs with age at blood collection > 6 months excluded in all analyses. ^2^ For 1 state, data for all screened newborns were obtained from live births posted by CDC, Natality online databases reporting counts of live births occurring within the United States to U.S. residents (https://wonder.cdc.gov/natality.html, accessed on 20 July 2020). ^3^ State-level variable. ^4^ Genotype information obtained by states includes variants detected by the state testing laboratory and those reported to the program after NBS. ^5^ CFTR2, Clinical and Functional Translation of *CFTR* (Cystic Fibrosis Transmembrane Conductance Regulator gene), Version 11 March 2019. (https://cftr2.org/, accessed on 30 December 2019).

**Table 2 IJNS-08-00058-t002:** Distribution of study factors by number of states in the analysis, study groups, and percentage of cystic fibrosis cases missed due to IRT being below program cutoffs.

			LB		CFTP		CFFN		CFFN/(CFTP + CFFN)
Characteristic	Response Categories	State N ^1^	N	%	N	%	N	%	% CF Cases Missed
Total	(Includes subjects with ≥1 missing values	13	11,246,522	100%	2019	100%	61	100%	2.93%
Demographic Factors									
Sex	Total ^2^	13	11,187,241	100%	2009	100%	61	100%	2.95%
	Female		5,463,266	49%	990	49%	30	49%	2.94%
	Male		5,723,975	51%	1019	51%	31	51%	2.95%
Race	Total ^2^	11	9,710,432	100%	1671	100%	55	100%	3.19%
	White		5,630,167	58%	1337	80%	40	73%	2.90%
	Black		1,381,196	14%	69	4%	7	13%	9.21%
	Total	11	9,708,228	100%	1671	100%	55	100%	3.19%
	Other		1,609,495	17%	137	8%	6	11%	4.20%
Ethnicity	Total ^2^	11	8,747,765	100%	1458	100%	54	100%	3.57%
	Hispanic		2,613,806	30%	262	18%	7	13%	2.60%
	Non-Hispanic		6,133,959	70%	1196	82%	47	87%	3.78%
Timing and Program Factors									
Day of	Total ^2^	11	9,338,370	100%	1614	100%	52	100%	3.01%
Week Birth	Sunday		918,843	10%	161	10%	3	6%	1.63%
Occurred	Monday		1,361,901	15%	205	13%	9	17%	3.57%
	Tuesday		1,532,034	16%	269	17%	4	8%	1.59%
	Wednesday		1,519,135	16%	246	15%	9	17%	3.75%
	Thursday		1,502,243	16%	274	17%	8	15%	2.71%
	Friday		1,474,005	16%	272	17%	8	15%	3.13%
	Saturday		1,030,209	11%	187	12%	11	21%	4.76%
Season of	Total ^1^	11	10,125,349	100%	1833	100%	55	100%	2.91%
Specimen	Winter		2,417,172	24%	422	23%	12	22%	2.76%
Collection	Spring		2,415,596	24%	435	24%	9	16%	2.03%
	Summer		2,585,070	26%	503	27%	17	31%	3.27%
	Fall		2,707,511	27%	473	26%	17	31%	3.47%
Maximum	Total	13	11,246,522	100%	2019	100%	61	100%	2.93%
consecutive	≥3		6,050,025	54%	993	49%	34	56%	3.31%
days lab closed	<3		5,196,497	46%	1026	51%	27	44%	2.56%
Program	Total	13	11,246,522	100%	2019	100%	61	100%	2.93%
Type	2-specimen		2,116,730	19%	367	18%	13	21%	3.42%
	1-specimen		9,129,792	81%	1652	82%	48	79%	2.82%
CF Algorithm Factors									
IRT	Total	13	11,246,522	100%	2019	100%	61	100%	2.93%
Cutoff	Fixed		6,444,732	57%	1025	51%	37	61%	3.48%
Type	Floating		4,801,790	43%	994	49%	24	39%	2.36%
IRT Cutoff	Total	13	11,246,522	100%	2019	100%	61	100%	2.93%
Level	<0.96		5,016,295	45%	921	46%	24	39%	2.54%
(percentile)	≥0.96		6,230,227	55%	1098	54%	37	61%	3.26%
	≥0.97		4,308,751	38%	679	34%	28	46%	3.96%
	≥0.98		4,054,073	36%	611	30%	23	38%	3.63%
	≥0.99		809,440	7%	126	6%	6	10%	4.55%
Biologic and Health Factors									
Genotype	Total ^2^	11	N/A	N/A	1853	100%	55	100%	2.88%
	Both variants unidentified				89	5%	3	5%	3.26%
	Both variants known CF-causing				1295	70%	27	49%	2.04%
	Other combinations				469	25%	25	45%	5.06%
Birth weight	Total ^2^	13	11,071,755	100%	1988	100%	61	100%	2.98%
(g)	<2500		917,081	8%	209	11%	8	13%	3.69%
	≥2500		10,154,674	92%	1779	89%	53	87%	2.89%
Gestational	Total ^2^	6	1,899,932	100%	426	100%	14	100%	3.18%
age (weeks)	<37		211,168	11%	50	12%	4	29%	7.41%
	≥37		1,688,764	89%	376	88%	10	71%	2.59%
Infant Feeding	Total ^2^	6	5,247,261	100%	824	100%	24	100%	2.83%
	Breast Only		2,727,016	52%	454	55%	9	38%	1.94%
	Any Formula Use		2,520,245	48%	370	45%	15	63%	3.90%
Meconium	Total ^2^	9	N/A	N/A	1090	100%	37	100%	3.28%
Ileus	Present				169	16%	11	30%	6.11%
	Not present				921	84%	26	70%	2.75%

Abbreviations: CFFN: false screen negative CF cases due to IRT below program cutoffs; CFTP: true screen positive CF infants; LB: all screened newborns; N/A: not available. ^1^ Analyses conducted on less than 13 states may be comprised of different states even though the number listed in the table may be the same. ^2^ Excludes missing or unknown values.

**Table 3 IJNS-08-00058-t003:** Number, percentage and odds ratios (90% confidence intervals) for study factors in CFTP, CFFN, CFFN ≥40 ng/mL and CFFN <40 ng/mL study groups.

		CFTP Cases		CFFN Cases								
				All		≥40 ng/mL		<40 ng/mL		Odds Ratios ^1^ (90% CI) ^2^		
Characteristic	Response Categories	N	%	N	%	N	%	N	%	All CFFNs	≥40 ng/mL	<40 ng/mL
Demographic Factors												
Sex	Total	2009	100%	61	100%	30	100%	31	100%			
	Female	990	49%	30	49%	14	47%	16	52%	1.0	0.9	1.1
	vs. Male	1019	51%	31	51%	16	53%	15	48%			
Race	Total	1671	100%	55	100%	26	100%	29	100%			
	White	1337	80%	40	73%	18	69%	22	76%	0.7	0.6	0.8
	vs. Rest	334	20%	15	27%	8	31%	7	24%			
	Black	69	4%	7	13%	4	15%	3	10%	3.4	4.2	2.7
	vs. Rest	1602	96%	48	87%	22	85%	26	90%	(1.7, 6.8)	(1.7, 10.6)	(0.96, 7.5)
	Other	137	8%	6	11%	3	12%	3	10%	1.4	1.4	1.5
	vs. Rest	1534	92%	49	89%	23	88%	26	90%			
Ethnicity	Total	1458	100%	54	100%	25	100%	29	100%			
	Hispanic	262	18%	7	13%	1	4%	6	21%	0.7	0.2	1.2
	vs. Rest	1196	82%	47	87%	24	96%	23	79%		(0.0, 0.98)	
Timing and Program Factors												
Day of	Total	1614	100%	52	100%	24	100%	28	100%			
Week Birth	Saturday	187	12%	11	21%	4	17%	7	25%	2.0	1.5	2.5
Occurred	vs. Other	1427	88%	41	79%	20	83%	21	75%	(1.2, 3.6)		(1.2, 5.3)
	Sunday	161	10%	3	6%	(a)	(a)	(a)	(a)	0.6	(a)	(a)
	vs. Other	1453	90%	49	94%	(a)		(a)				
	Tuesday	269	17%	4	8%	2	8%	2	7%	0.4	0.5	0.4
	vs. Other	1345	83%	48	92%	22	92%	26	93%	(0.2, 0.99)	(0.1, 1.5)	(0.1, 1.3)
Season of	Total	1833	100%	55	100%	25	100%	30	100%			
Specimen	Summer/Fall	976	53%	34	62%	13	52%	21	70%	1.4	1.0	2.0
Collection	vs. Other	857	47%	21	38%	12	48%	9	30%			(1.1, 4.0)
Maximum Con-	Total	2019	100%	61	100%	30	100%	31	100%			
secutive Days	≥3	993	49%	34	56%	13	43%	21	68%	1.2	0.7	2.0
Lab Closed	<3	1026	51%	27	44%	17	57%	10	32%			(1.1, 3.8)
Program	Total	2019	100%	61	100%	30	100%	31	100%			
Type	2-specimen	367	18%	13	21%	7	23%	6	19%	1.2	1.4	1.1
	vs. 1-specimen	1652	82%	48	79%	23	77%	25	81%			
CF Algorithm Factors												
IRT	Total	2019	100%	61	100%	30	100%	31	100%			
Cutoff	Fixed	1025	51%	37	61%	20	67%	17	55%	1.5	2.0	1.2
Type	vs. Floating	994	49%	24	39%	10	33%	14	45%		(1.1, 3.8)	
IRT Cutoff	Total	2019	100%	61	100%	30	100%	31	100%			
Level	≥0.96	1098	54%	37	61%	20	67%	17	55%	1.3	1.7	1.0
(percentile)	vs. <0.96	921	46%	24	39%	10	33%	14	45%			
Biologic and Health Factors												
Genotype	Total	1853	100%	55	100%	25	100%	30	100%			
	Both variants unidentified	89	5%	3	5%	1	4%	2	7%	1.1	0.8	1.4
	vs. other genotypes	1764	95%	52	95%	24	96%	28	93%			
	Both variants CF-causing	1295	70%	27	49%	9	36%	18	60%	0.4	0.2	0.6
	vs. other genotypes	558	30%	28	51%	16	64%	12	30%	(0.3, 0.7)	(0.1, 0.5)	
	Other combinations	469	25%	25	45%	15	60%	10	33%	2.5	4.4	1.5
	vs. other genotypes	1384	75%	30	55%	10	40%	20	67%	(1.6, 3.9)	(2.2, 8.7)	
Birth weight	Total	1988	100%	61	100%	30	100%	31	100%			
(g)	<2500	209	11%	8	13%	3	10%	5	16%	1.3	0.9	1.6
	≥2500	1779		53		27		26				
Gestational age	Total	426	100%	14	100%	8	100%	6	100%			
(weeks)	<37	50	12%	4	29%	1	13%	3	50%	3.0	1.1	7.5
	≥37	376	88%	10	71%	7	67%	3	50%	(1.1, 8.2)		(1.9, 29.5)
Infant Feeding	Total	824	100%	24	100%	13	100%	11	100%			
	Any Formula Use	370	45%	15	63%	5	38%	10	91%	2.0	0.8	12.3
	Breast Only	454	55%	9	37%	8	62%	1	9%	(1.01, 4.1)		(2.2, 69.1)
Meconium	Total	1090	100%	37	100%	21	100%	16	100%			
Ileus	Present	169	16%	11	30%	7	33%	4	25%	2.3	2.7	1.8
	Not present	921	84%	26	70%	14	67%	12	75%	(1.2, 4.2)	(1.3, 5.9)	

Abbreviations: CFFN: false screen negative CF cases due to IRT below program cutoffs; CFTP: true screen positive CF infants; CI: confidence interval; (a): cell sizes too small to present results. ^1^ 90% CIs are presented when OR ≤ 0.5 or ≥2.0. ^2^ Compared to odds in CFTP cases.

## Data Availability

The information collected from state CF newborn screening programs was obtained under conditions that the data would not be made publicly available or stratified by state. In order to assist future researchers interested in replicating this study, we have made our survey forms and instructions available as Appendix A online.

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
