# Peer review of "Missed Cystic Fibrosis Newborn Screening Cases due to Immunoreactive Trypsinogen Levels below Program Cutoffs: A National Survey of Risk Factors"

_2409-515X, 2022, doi:10.3390/ijns8040058_

Round 1

Reviewer 1 Report

This manuscript is the result of a tremendous effort to find causes of low IRT levels in dried blood samples which asked input from many states in the USA. Considered by the authors as only an exploratory study it shows some unexpected findings, such as a higher risk of low IRT levels in formula fed newborns but also confirms earlier findings. The result and recommendations of the study seem worthwhile.

Author Response

Thank you for your gracious comments.

Reviewer 2 Report

It is well known that all (CF) screening programs miss cases and this is a great attempt to identify factors which will inform efforts to minimise false negative screening results. The authors are to be commended for their efforts to collect and analyse a lot of comparable data from a number of sources. Some aspects merit further discussion or acknowledgement in the text.

1 Not all CF is the same, and it is more likely that pancreatic sufficient cases will be missed due to IRT not raised. Is there data available to support this? Continuing the hypothesis that missed cases are likely to be milder, it would be interesting to know if any clinical details are available on the missed cases identified before 28d of age as with such early identification they may be quite severe.

2 In the public health context population screening is not considered appropriate for persons at increased prior risk of the disorder – they should be referred directly for diagnostic workup – in this case meconium ileus and family history might exclude infants from screening and hence from inclusion in an analysis of false negative results (which doesn’t preclude a newborn screening card being collected and tested for other disorders on the panel, or CF, it is more about the screening metrics).

3 Are there any reasons known why there is a difference in race and ethnicity in false negative results? Is it possible to speculate? Would the authors suggest differential cutoffs (same MoM or centile for each group which would presumably be different numerically)?

4 It is unlikely a lab closed for 4 days now and again vs one closed for 3 makes a difference in degradation of IRT – more likely is the total delay between specimen collection and testing (this may also at least partly account for the day of the week difference observed). Is it possible to analyse for the delay in testing as a variable? However, as the authors note in the USA there have been significant moves to reporting of newborn screening results at an earlier age (labs open more days, samples not batched, couriered every day).

5 The importance of developing a test to check for enzyme integrity as a criterion for sample acceptability is increasing as more disorders are added to screening panels – IRT, GALT, biotinidase, lysosomal enzymes to start with.
